# A Peritoneal Purse-String Suture Prevents Symptomatic Lymphoceles in Retzius-Sparing Robot-Assisted Radical Prostatectomy

**DOI:** 10.3390/jcm12030791

**Published:** 2023-01-19

**Authors:** Niklas Harland, Mohammed Alfarra, Eva Erne, Moritz Maas, Bastian Amend, Jens Bedke, Arnulf Stenzl

**Affiliations:** Department of Urology, Medical Faculty and University Hospital, Eberhard-Karls-University Tuebingen, 72076 Tuebingen, Germany

**Keywords:** prostate cancer, prostatectomy, lymphocele, robot-assisted

## Abstract

Background: The retzius-sparing approach for robotic-assisted radical prostatectomy (RARP) has been increasingly adopted. Symptomatic lymphoceles are a widespread complication of RARP with pelvic lymph node dissection. Here, we present a new technique, the peritoneal purse-string suture (PPSS), that seems to reduce the rate of symptomatic lymphoceles following retzius-sparing RARP with extended pelvic lymph node dissection (ePLND). Methods: The radical prostatectomy and bilateral lymphadenectomy are performed through three separate peritoneal openings. The PPSS uses a single suture in a way similar to a purse-string suture; the openings of both lymphadenectomy fields are widened, and the rectovesical opening from the prostatectomy is simultaneously closed. This report retrospectively evaluates the perioperative and postoperative outcomes of two consecutive patient cohorts undergoing RARP with ePLND by a single surgeon between May 2015 and June 2019, one cohort prior to introducing PPSS as control (*n* = 145) and the other after introducing PPSS (*n* = 91). Results: The two study groups were comparable on baseline characteristics, except ASA. There were no Clavien–Dindo grade IV-V complications, and comparable rates of grade I-III complications. The difference in postoperative lymphocele formation was 22% in PPSS versus 27% in the control group (*p* = 0.33). The rate of symptomatic lymphoceles was significantly lower in the PPSS group (3% vs. 10%, *p* = 0.047). Conclusion: The PPSS is a feasible procedure that reduces symptomatic lymphoceles in patients undergoing RARP with a retzius-sparing approach.

## 1. Introduction

The robot-assisted laparoscopic approach to radical prostatectomy has been widely adopted over the last decade. It is indisputable that it comes with the same advantages as conventional laparoscopy, namely reduced blood loss and expedited recovery [1]. The improvements in oncological safety, postoperative-continence, and erectile function are disputed [2]. Most surgical techniques in robot-assisted radical prostatectomy (RARP) copy the approach of open surgery, with anterior release of the bladder and subsequent preparation of the prostate.

Recently, a new technique has been described, with preparation from the posterior part of the prostate, sparing the spatium retzii [3]. This keeps intact part of the autonomous nerves running ventrally and also keeps the anterior wall of the bladder, including the bladder neck, intact. Initial results suggested that the time to continence is thereby reduced [4,5,6].

Lymphoceles represent one of the most common complications of radical prostatectomy with pelvic lymphadenectomy [7]. Among other problems, they can lead to infections, thrombosis, and pain. Therefore, percutaneous aspiration, placement of drainage, or surgical fenestration may be necessary to deal with symptomatic lymphoceles [8]. Some approaches to prevent lymphoceles in radical prostatectomy with pelvic lymphadenectomy have been described for open surgery and RARP, yet the outcomes have been variable [9,10]. Porpiglia et al. reported a reduced rate of symptomatic lymphoceles using a laparoscopic transperitoneal approach vs. an extraperitoneal approach [11]. In order to further increase the peritoneal surface with contact to the iliac fossa, some groups have established methods to reconfigure a peritoneal flap [9,12,13]. Other groups have investigated the application of hemostatic agents [14,15] or different sealing modalities applied to the lymphatic vessels [16,17] in order to reduce the rate of lymphoceles following extended pelvic lymph node dissection (ePLND).

This paper presents an approach for the prevention of symptomatic lymphoceles following retzius-sparing radical prostatectomy with ePLND, and reports our initial results regarding its feasibility and effectiveness.

## 2. Materials and Methods

### 2.1. Study Design and Patients

This retrospective study included 236 consecutive patients undergoing RARP with ePLND with a retzius-sparing approach, by a single surgeon between May 2015 and June 2019. The peritoneal purse-string suture (PPSS) was introduced in our department in December 2017, and is consistently performed for every patient undergoing RARP with ePLND.

### 2.2. Data Collection

We retrospectively collected demographic and baseline medical characteristics, oncological characteristics, intraoperative surgical data, and postoperative outcomes data. The presence of lymphoceles was evaluated by routine ultrasound of the abdomen and pelvis between 5 and 7 days postoperatively, or by CT scan for diverse indications (pain, fever, etc.). Lymphoceles was identified as any postoperative fluid collection in the pelvis or lower abdomen detected by ultrasound or CT. They were defined as asymptomatic if there were no concomitant related symptoms; they were defined as symptomatic if they were found in a patient with fever, pain, urinary retention, or swelling of one or both lower extremities. 

### 2.3. Surgical Procedure

All patients underwent RARP with ePLND, performed by a single surgeon with more than 10 years of experience in RARP. RARP was performed according to the retzius-sparing approach first described by Galfano et al. [3]. The DaVinci Si surgical system was used until December 2018, and the DaVinci Xi surgical system was used thereafter. Before December 2017, all surgeries were performed without the PPSS (about 600 such RARPs). Since December 2017, all patients have received the PPSS at the end of the procedure. Nothing else was changed in the rest of the surgical technique.

During retzius-sparing RARP, the preparation of the prostate and vesico-urethral anastomosis is performed through a small incision subtrigonally, ventral to the rectum. To enable pelvic lymphadenectomy, two separate incisions to the peritoneum must be made—one on each side of the pelvis. Lymphostasis was performed with clips.

During the PPSS, the medial lip of the lateral incisions is attached to the posterior incision. This enables continuous drainage of possible lymphatic discharge to the peritoneal cavity. At the same time, the posterior incision is closed to prevent postoperative leakage from the anastomosis (Figure 1). 

### 2.4. Statistical Analysis

Descriptive data are presented as the median with interquartile range. Wilcoxon and Χ2 tests were used to compare the two study groups’ results. Statistical significance was defined as *p* < 0.05. The statistical analysis was performed using JMP version 15.0.

## 3. Results

### 3.1. Patient Characteristics

The study included 91 patients receiving PPSS, and 145 without that additional suture. The two groups did not differ in age, BMI, prostate volume, or previous abdominal surgeries (Table 1). More patients were Grade 2 according to the American Society of Anesthesiologists physical status classification system (ASA) in the intervention group (80% vs. 59%). The two groups were comparable in preoperative PSA-level, International Society of Urological Pathology (ISUP) group upon biopsy, and clinical tumor stage; consequently, they were also comparable in risk group according to D’Amico (Table 1).

### 3.2. Intraoperative Parameters

The rate of complete nerve sparing was comparable between both groups. The median duration of the surgery was not statistically significant between control and PPSS group (207 min vs. 196 min). The median lymph node yield was 13 in the whole collective of both groups (*p* = 0.94), with positive lymphnodes in 3 patients (3%) in the control group and 5 patients (3%) in the PPSS group (*p* = 0.96). The rate of intraoperative complications was not statistically significantly different between the two study groups. No patient experienced an intraoperative complication above Grade III according to the Clavien–Dindo classification [18].

### 3.3. Postoperative Outcomes

During postoperative surveillance, there was no significant difference between the two groups in the time to removal of the catheter, the decrease in hemoglobin levels, bowel movements, or complications within 30 days, classified according to Clavien and Dindo (Table 2).

The rate of symptomatic lymphoceles was significantly lower in the PPSS group compared to the control group (3 (2.7%) vs. 15 (10.3%), *p* = 0.047). The difference in the rate of asymptomatic lymphoceles between the PPSS group and the control group was not statistically significant (20 (22%) vs. 40 28%, *p* = 0.3). The symptoms manifested with pain (10/18), signs of infection (7/18), leg oedema (4/18) and urinary retention (1/18). Patients with symptomatic lymphoceles received percutaneous aspiration (1/18), percutaneous drainage (15/18), laparoscopic fenestration (4/18), and transurethral catheterization (3/18).

The yield of lymph nodes was not significantly associated with the occurrence of lymphoceles (asymptomatic lymphocele *p* = 0.052; symptomatic lymphocele *p* = 0.06). Body mass index (BMI), duration of surgery, D’Amico risk group, estimated blood loss, and previous abdominal surgeries were not significantly associated with either asymptomatic or symptomatic lymphocele formation. 

## 4. Discussion

Extended pelvic lymphadenectomy is recommended by the guidelines for patients with a risk of nodal metastasis > 5% [19]. While a recent meta-analysis did not show an oncological benefit of ePLND [20], its diagnostic value for staging over the existing imaging techniques justifies its recommendation [21].

This advantage comes with a relevant risk for complications, the most frequent being the formation of lymphoceles. The rate of lymphoceles reported in the literature varies between 4% and 48% [17,22] most likely depending on the mode of follow-up. Most cases are asymptomatic and do not require an intervention. Only 3% to 5% of patients undergoing ePLND have a symptomatic lymphocele, which is drained or fenestrated surgically [17,22]. Many groups have presented approaches to reduce the rate of symptomatic lymphoceles for open or laparoscopic PLND. Although retrospective studies have reported a significant reduction in the formation of symptomatic lymphoceles [9], prospective trials have found no differences [23]. 

The retzius-sparing approach to robotic-assisted radical prostatectomy was first introduced by Galfano et al. in 2010 [3]. Since then, many groups have reported on the advantages of this technique for functional outcomes [4]. The surgical approach of retzius-sparing RARP compared to the anterior approach was a significant predictor for the occurrence of symptomatic lymphoceles, with an odds ratio of 23.8 [24]. This could be due to the fact that, compared to the anterior approach, RARP makes small incisions to the peritoneal cavity. While this higher risk is being debated [25], we describe here the first technique to reduce symptomatic lymphoceles in retzius-sparing RARP.

This study has some limitations. Foremost, it is a retrospective study, which means that the original data collection was not deliberately designed and performed to address the study questions as it would be for a prospective study. Second, it is a single-surgeon study, so the patient sample might not be representative of the larger patient population, and the surgeon’s relative performance of the two techniques may not be comparable to the relative performance of the two techniques by other surgeons. Third, the two study groups were from different time periods, and in each timeframe only one approach, with or without PPSS, was performed for all patients. Thus, the patient groups are constrained within the time period, so apparent differences in outcomes between the two surgical techniques might be due instead to differences between those two time periods with regard to patient selection, their general healthcare, or other factors. Fourth, all patients were evaluated for asymptomatic lymphoceles through ultrasound scan, which may not have detected all lymphoceles. Although computed tomography has higher sensitivity, that advantage must be weighed against the exposure to radiation and contrast-agents and higher costs that come with the procedure. In any case, the difference between the two imaging modalities for identifying symptomatic lymphoceles is most likely negligible. 

## 5. Conclusions

PPSS is a feasible and effective procedure to prevent symptomatic lymphocele formation in patients undergoing retzius-sparing RARP with ePLND. The fenestration of the iliac fossa prevents early closure and thereby enables persistent drainage of lymph fluid. These results must be confirmed in a prospective randomized trial.

## Figures and Tables

**Figure 1 jcm-12-00791-f001:**
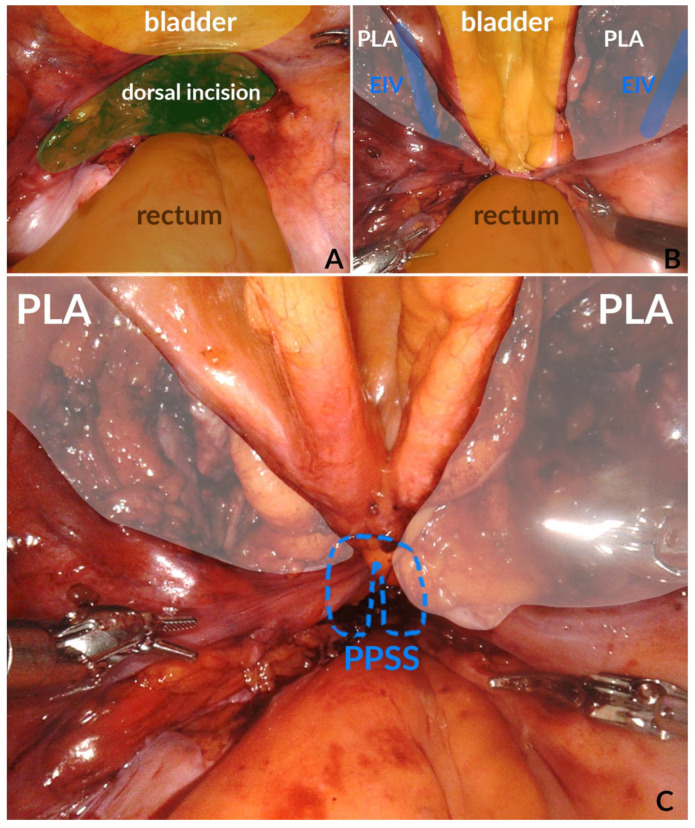
(**A**) Subtrigonal access to the prostatic bed through dorsal incision; (**B**) iliac fossa with pelvic lymphadenectomy on both sides; (**C**) opening of the iliac fossa on both sides and closure of the dorsal incision through a peritoneal purse-string suture. Abbreviations: PPSS: peritoneal purse string suture, EIV: external iliac vein, PLA: pelvic lymphadenectomy.

**Table 1 jcm-12-00791-t001:** Patient characteristics.

				Control (*n* = 145)	PPSS (*n* = 91)	
Age: median (IQR)				65 (59–71)	66 (59–71)	0.9
BMI median (IQR) kg/m²				26.5 (24.15–28.4)	25.6 (23.9–28)	0.2
ASA-score	I			52 (36%)	11 (12%)	<0.001
	II			85 (59%)	73 (80%)	
	III			8 (5%)	7 (8%)	
	IV–VI			0	0	
Preoperative PSA: median (IQR) ng/mL				6.60 (5.27–8.57)	5.74 (4.73–8.35)	0.1
D’Amico risk group			Low	12 (8%)	8 (9%)	0.9
			Intermediate	103 (71%)	65 (71%)	
			High	30 (21%)	18 (20%)	
Previous abdominal surgery				85 (59%)	57 (65%)	0.3
Prostate volume (IQR) ml				40 (30–50)	35 (30–48)	0.1
ISUP group		1		22 (15%)	9 (10%)	0.8
		2		63 (43%)	41 (45%)	
		3		41 (28%)	27 (30%)	
		4		13 (9%)	11 (12%)	
		5		6 (4%)	3 (3%)	
Clinical tumor stage		1		89 (61%)	59 (65%)	0.4
		2		56 (39%)	32 (35%)	

Abbreviations: IQR: interquartile range, PPSS: peritoneal purse-string suture.

**Table 2 jcm-12-00791-t002:** Clinical outcomes.

		Control	PPSS	*p*
Catheterization: median (IQR) days		5 (4–24)	5 (5–9)	0.7
Postoperative ISUP	1	8 (6%)	1 (1%)	0.3
	2	68 (47%)	46 (51%)	
	3	50 (34%)	32 (35%)	
	4	7 (5%)	7 (8%)	
	5	12 (8%)	5 (5%)	
Estimated blood loss: median (IQR) mL		350 (225–400)	400 (300–400)	0.057
Reduction in hemoglobin on 1st postoperative day: median (IQR) g/dL		2.3 (1.8–2.8)	2.1 (1.6–2.7)	0.4
Postoperative complication grade according to Clavien–Dindo	0	111 (77%)	72 (79%)	0.6
	1	3 (2%)	4 (4%)	
	2	13 (9%)	5 (5%)	
	3	18 (12%)	10 (12%)	
	4–5	0	0	
Duration of surgery median (IQR) min		207 (178.5–226.5)	196 (172–224)	0.08

Abbreviations: IQR: interquartile range, ISUP: International Society of Urological Pathology; PPSS: peritoneal purse-string suture.

## Data Availability

Not applicable.

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
