# Peer review of "A Peritoneal Purse-String Suture Prevents Symptomatic Lymphoceles in Retzius-Sparing Robot-Assisted Radical Prostatectomy"

_jcm, 2023, doi:10.3390/jcm12030791_

Round 1
Reviewer 1 Report
Lymphocele is a known complication following pelvic lymph nodes dissection during RALP. In the manuscript the authors described a new technique with purse-string suture during RS-RALP aiming to reduced lymphocele occurrence. Using serial data from a single surgeon, the authors showed that purse-string suture can reduced formation of symptomatic lymphocele with a statistically significant difference. Despite some limitations as mentioned by authors in the manuscript, the data presented support their conclusions.
Some comments
1. Lymphoceles were detected using a combination of routine US and as needed CT per authors. Can you please provide the breakdown of how many cases were detected by each imaging modality divided by symptomatic vs asymptomatic lymphoceles?
2. Was a surgical drain placed? If so, did it affect the occurrence of lymphoceles?
3. Line 130-132 seems to be editorial from authors. Please revise accordingly.
Author Response
Dear Reviewer 1,
Thank you for considering our manuscript and letting us resubmit a second draft for publication. We appreciate the time and effort that you dedicated to providing feedback and are very grateful for your comments.
Please see a point-by-point response to the referees’ comments and concerns below.
- Lymphoceles were detected using a combination of routine US and as needed CT per authors. Can you please provide the breakdown of how many cases were detected by each imaging modality divided by symptomatic vs asymptomatic lymphoceles?
Answer: Ultrasound was the routine diagnostic tool and was performed by nearly all patients as first imaging. CT scan were only used as primary diagnostic in 2 cases. Because of the low number of primary CT scans (n = 2) further statistical analysis was deferred.
- Was a surgical drain placed? If so, did it affect the occurrence of lymphoceles?
Answer: No. The placement of drainage is not part of our surgical routine.
- Line 130-132 seems to be editorial from authors. Please revise accordingly.
Answer: Is changed. Thank you for your hind.
We hope that the point-by-point response will remedy your concerns, but we are happy to consider further revisions, and we thank you for your continued interest in our research.
Yours sincerely,
Eva Erne
Reviewer 2 Report
Reviewer´s comments on A Peritoneal Purse-String Suture Prevents Symptomatic Lymphoceles in Retzius-Sparing Robot-Assisted Radical Prostatectomy.
Overall I find nothing to declare in respect of appropriateness of context and purpose of study reflected by the abstract and the introduction. The methods used are valid, underlined by previous publications and the results are correctly presented in good quality figures and tables. The discussion and conclusions are supported by the data.
Finally, I come to the conclusion that the authors provide interesting data.
Author Response
Dear Reviewer 2,
thank you very much for your comment. We are pleased about your interest in our research.
Yours sincerely,
Eva Erne